# Effects of therapeutic hypothermia on death among asphyxiated neonates with hypoxic-ischemic encephalopathy: A systematic review and meta-analysis of randomized control trials

Biruk Beletew Abate[1]*, Melaku Bimerew[1], Bereket Gebremichael[2], Ayelign Mengesha Kassie[1‡], MesfinWudu Kassaw[1‡], Teshome Gebremeskel[1‡], Wubet Alebachew Bayih[3‡]

1 Department of Nursing, College of Health Sciences, Woldia University, Woldia, Ethiopia, 2 College of Health Sciences, Addis Ababa University, Addis Ababa, Ethiopia, 3 College of Health Sciences, Debre Tabor University, Debre Tabor, Ethiopia

☯ These authors contributed equally to this work.
‡ AMK, MWK, TG and WAB also contributed equally to this work.
* birukkelemb@gmail.com

## Abstract

### Background

Hypoxic perinatal brain injury is caused by lack of oxygen to baby's brain and can lead to death or permanent brain damage. However, the effectiveness of therapeutic hypothermia in birth asphyxiated infants with encephalopathy is uncertain. This systematic review and meta-analysis was aimed to estimate the pooled relative risk of mortality among birth asphyxiated neonates with hypoxic-ischemic encephalopathy in a global context.

### Methods

We used the Preferred Reporting Items for Systematic Review and Meta-Analysis (PRISMA) guidelines to search randomized control trials from electronic databases (PubMed, Cochrane library, Google Scholar, MEDLINE, Embase, Scopus, Web of Science, Cochrane Central Register of Controlled Trials (CENTRAL), and meta register of Current Controlled Trials (mCRT)). The authors extracted the author's name, year of publication, country, method of cooling, the severity of encephalopathy, the sample size in the hypothermic, and non-hypothermic groups, and the number of deaths in the intervention and control groups. A weighted inverse variance fixed-effects model was used to estimate the pooled relative risk of mortality. The subgroup analysis was done by economic classification of countries, methods of cooling, and cooling devices. Publication bias was assessed with a funnel plot and Eggers test. A sensitivity analysis was also done.

**Data Availability Statement:** All relevant data are within the paper and its Supporting information files.

**Funding:** The author(s) received no specific funding for this work.

**Competing interests:** The authors have declared that no competing interests exist.

**Abbreviations:** HIE, Hypoxic-ischemic encephalopathy; WHO, World Health Organization; CI, Confidence interval; AOR, Adjusted Odds Ratio; BW, Birth Weight; GA, Gestational Age; RR, Relative Risk; PRISMA, Preferred Reporting Items for Systematic Reviews and Meta-Analyses; SHC, Selective Head Cooling; WBC, Whole Body Cooling; LMI, Low and middle income.

## Results

A total of 28 randomized control trials with a total sample of 35, 92 (1832 hypothermic 1760 non-hypothermic) patients with hypoxic-ischemic encephalopathy were used for the analysis. The pooled relative risk of mortality after implementation of therapeutic hypothermia was found to be 0.74 (95%CI; 0.67, 0.80; $I^2$ = 0.0%; p<0.996). The subgroup analysis revealed that the pooled relative risk of mortality in low, low middle, upper-middle and high income countries was 0.32 (95%CI; -0.95, 1.60; $I^2$ = 0.0%; p<0.813), 0.5 (95%CI; 0.14, 0.86; $I^2$ = 0.0%; p<0.998), 0.62 (95%CI; 0.41–0.83; $I^2$ = 0.0%; p<0.634) and 0.76 (95%CI; 0.69–0.83; $I^2$ = 0.0%; p<0.975) respectively. The relative risk of mortality was the same in selective head cooling and whole-body cooling method which was 0.74. Regarding the cooling device, the pooled relative risk of mortality is the same between the cooling cap and cooling blanket (0.74). However, it is slightly lower (0.73) in a cold gel pack.

## Conclusions

Therapeutic hypothermia reduces the risk of death in neonates with moderate to severe hypoxic-ischemic encephalopathy. Both selective head cooling and whole-body cooling method are effective in reducing the mortality of infants with this condition. Moreover, low income countries benefit the most from the therapy. Therefore, health professionals should consider offering therapeutic hypothermia as part of routine clinical care to newborns with hypoxic-ischemic encephalopathy especially in low-income countries.

## Introduction

Hypoxic-ischemic encephalopathy (HIE) is a complication resulting from intrapartum and neonatal asphyxia. Adverse intrapartum events remains a major cause of neonatal mortality and burden of disease in emerging economies [1, 2]. Neonatal encephalopathy due to perinatal asphyxia occurs in 1 up to 3 per 1000 live births in high-income countries, and in up to 20 per 1000 live births in low and middle-income countries [3]. The burden in low and middle-income countries is far higher than in high-income countries, and it accounts for approximately one million deaths annually [4]. If not treated, 62% of infants with perinatal hypoxic brain injury will die or have moderate to severe disabilities by the age of 18 to 22 months; treatment reduces this rate to 41% [5, 6]. Survivors also develop long-term neurologic disabilities as follows: 45% have cognitive and developmental delay or learning difficulties, 29%, some degree of cerebral palsy, 26%, blindness or vision defects, 17%, gross motor and coordination problems, epilepsy, 9%, hearing loss or deafness, and 1%, behavioral issues [7, 8].

Intrapartum hypoxia resulting in hypoxic-ischemic encephalopathy (HIE) is one of the causes of neonatal encephalopathy, with no definitive test to make the diagnosis. In addition, very little knowledge is available in terms of neuroprotective strategies, the use of therapeutic hypothermia (TH) is one of the strategies commonly used and shown most promising neuroprotective intervention [9, 10].

In neonates with perinatal asphyxia, admission hyperoxemia increased the incidence of Hypoxic Ischemic Encephalophaty (HIE). Among neonates with HIE, admission hyperoxemia increased the risk of abnormal brain magnetic resonance imaging findings. The careful use of oxygen during and after resuscitation is necessary [11]. The phenomenon in which

oxygen supplementation following a period of oxygen deficiency augments the injury is known as "the oxygen paradox". Thus a powerful mean to reduce HIE is to avoid hyperoxia which results in rapid cell swelling [12].

Regarding the percentage of oxygen a systematic review and meta-analysis revealed that there is a significant reduction in the risk of neonatal mortality and a trend towards a reduction in the risk of sever HIE in newborn resuscitated with 21% O2 [13]. Hypoxic insults to the brain have been associated with an elevation in brain temperature. It is speculated that this temperature increase is caused by increased metabolic demands and inflammatory mediators released after acute ischemic injury [14]. Hypothermia prevents death in neonates with hypoxic-ischemic encephalopathy due to perinatal asphyxia and considered to be the standard treatment for infants with this condition [1, 15]. It has been suggested that lowering core body temperature by 1˚C results in a 6% to 10% reduction in whole-body metabolic demands [16].

Two methods of hypothermia are commonly used (selective head cooling and whole-body hypothermia) [17]. Brain cooling is effective in reducing the extent of brain injury even when it is initiated up to 5.5 hours after brain ischemia in near-term sheep fetuses. Reductions in brain temperature by 2˚C to 5˚C provide neuroprotection in newborn and adult animal models of brain ischemia [18].

Improved survival and neurodevelopmental outcome at 18 months of age have been reported in multiple trials of therapeutic hypothermia, and currently, it is the only neuroprotective strategy for neonates suspected to have suffered an intrapartum hypoxic-ischemic event.

According to International Liaison Committee on Resuscitation (ILCOR) 2020, hypothermia treatment cooling only be considered in neonatal care facilities with the capabilities for multidisciplinary care and availability of adequate resources to offer intravenous therapy, respiratory support, pulseoximetry, antibiotics, anticonvulsants, and pathology testing [19].

Previous randomized control trials conducted across the world reported the relative risk of mortality among birth asphyxiated neonates with hypoxic-ischemic encephalopathy after the implementation of therapeutic hypothermia. The relative risk of mortality in such randomized control trials ranged from 0.00 [20] to 0.95 [21]. This indicates, there is an inconsistency report on the relative risk of mortality across different countries in the world. Moreover, there is no globally denoted pooled data which can be used as a baseline in designing strategies for the prevention of neonatal mortality particularly due to hypoxic-ischemic encephalopathy. Therefore, this systematic review and meta-analysis aimed to estimate the pooled relative risk of mortality among birth asphyxiated neonates with hypoxic-ischemic encephalopathy in a global context.

## Methods

### Reporting

The results of this review were reported based on the Preferred Reporting Items for Systematic Review and Meta-Analysis statement (PRISMA) guideline (S1 Checklist).

### Searching strategy and information sources

We identified studies providing data on the effect of therapeutic hypothermia/ cooling therapy on newborn mortality from hypoxic-ischemic encephalopathy from PubMed, Cochrane library, Google Scholar, MEDLINE, Embase, Scopus, Web of Science, Cochrane Central Register of Controlled Trials (CENTRAL), and metaRegister of Current Controlled Trials (mCRT). The last search was performed in April, 2020. The search was performed by using keywords/ phrases and medical subject headings (MeSH) terms. To retrieve additional potentially eligible

studies snowball searching in the reference list of papers was also conducted. Articles with incomplete reported data were handled through contacting corresponding authors. We used the search terms independently and/or in combination using Boolean operators like "OR" or "AND".

The core search terms and phrases were "neonates", "newborn", "infant", and "therapeutic hypothermia", "cooling therapy", "asphyxia", "hypoxic-ischemic", "hypoxic-ischemic", "encephalopathy". The search strategies were developed using different Boolean operators. Remarkably, to fit advanced PubMed database, the following search strategy was applied: (neonate [MeSH Terms] OR newborn OR perinatal OR infant) AND (hypothermia [MeSH Terms] OR cool OR cooling OR temperature OR body temperature) AND (death OR mortality) AND (asphyxia [MeSH Terms] OR hypoxic-ischemic OR hypoxic-ischaemic OR hypoxia OR brain OR encephalopathy AND therapy (S1 Table).

## Study selection

Retrieved studies were exported to reference manager software, Endnote version 8 to remove duplicate studies. Two investigators (BB and TG) independently reviewed the retrieved studies using their titles and abstracts before retrieval of full-text papers. We used pre-specified inclusion criteria to further screen the full-text articles. Disagreements were discussed during a consensus meeting with other reviewers (AM and MW) for the final selection of studies to be included in the analysis.

## Eligibility criteria

We included RCTs that analyzed the effect of whole-body hypothermia or selective head cooling on newborn mortality from hypoxic-ischemic encephalopathy compared with non-hypothermic patients. We only included studies that fulfilled all criteria. There were no restrictions for language, length of follow up, publication date, or status. Researches that did not report our outcome of interest were excluded.

## Quality assessment

After combining the Database search results duplicate articles were removed using Endnote (version X8). The Joanna Briggs Institute (JBI) Critical Appraisal Checklist for Randomized Controlled Trials was used [22, 23]. Four independent authors appraised the quality of all potential studies to be included for analysis. The appraisal was repeated by exchanging with each other. Thus, one paper was appraised by two Authors. Any disagreement between the reviewers was solved by taking the mean score of the two reviewers (S2 Table).

## Data extraction

The authors developed a data extraction form on the excel sheet which includes the author's name, year of publication, country, method of cooling, the severity of encephalopathy, the sample size in the hypothermic, and non-hypothermic groups, and the number of deaths in the intervention and control groups. The data extraction sheet was piloted using 4 papers randomly. The extraction form was adjusted after piloted the template. Two of the authors extracted the data using the extraction form in collaboration. The third and fourth authors checked the correctness of the data independently. Any disagreements between reviewers were resolved through discussions with a third reviewer and fourth reviewer if required. Any mistyping of data was resolved through crosschecking with the included papers. If we got

incomplete data, we excluded the study after two attempts were made to contact the corresponding author by email.

### Statistical analysis

The primary outcome was the relative risk (RR) of death, which was calculated as the proportion of death among hypothermic over the proportion of death among non-hypothermic patients.

After the data was extracted using Microsoft Excel format, we imported the data to STATA version 14.0 statistical software for further analysis. Using the binomial distribution formula, standard error was calculated for each study. We pooled the estimates RR of death by a fixed-effects model meta-analysis [24]. The pooled estimates RR of death with 95% CI were presented using forest plots. We examined the heterogeneity between the studies using Cochrane's Q statistics (Chi-square), inverse variance (I2), and p-values [25]. Subgroup analysis was done by stratifying studies using the method of cooling and the economy classification of the country where the study was conducted (low-income economies, lower-middle-income economies, upper-middle-income economies, and high-income economies) [26].

When statistical pooling is not possible, non-pooled data was presented in table form. Sensitivity analysis was employed to see the effect of a single study on the overall estimation. Publication bias was checked by the funnel plot and more objectively through Egger's regression test [27].

## Results

### Study selection

A total of 21,572 studies were identified using electronic searches (through Database searching (n = 21,560) and other sources (n = 12)). After duplication removed, a total of 11,150 articles remained (10422 duplicated). Finally, 1500 studies were screened for full-text review and, 28 articles with a total sample of 3.592 (1,832 hypothermic 1,760 non-hypothermic) patients were included for the final analysis (Fig 1).

### Characteristics of included studies

Table 1 summarizes the characteristics of the 28 included studies in the systematic review and meta-analysis [20, 21, 28–50]. Regarding the income of countries in which the trail was done, 15 studies were done in high income, 4 studies in upper middle income, 7 in low middle income, and 2 studies in low-income countries. Regarding methods of cooling used, 20 studies used whole body cooling while the remaining 8 studies used selective head cooling. Eleven included studies used cold gel pack, 9 studies used cooling blanket, and 8 studies used cooling caps as a cooling device. There were 1126/3592 deaths, 483/1832 in the hypothermia group, and 643/1760 in the control group (Tables 1 and 2).

### Characteristics of excluded studies

Almost all excluded studies were case series. The majority used whole body cooling methods. The highest proportion of those studies was from Africa. Regarding the exclusion criteria some studies were excluded because the study discontinued due to adverse outcomes, study details unclear, the protocol only, and they were case series (Table 3).

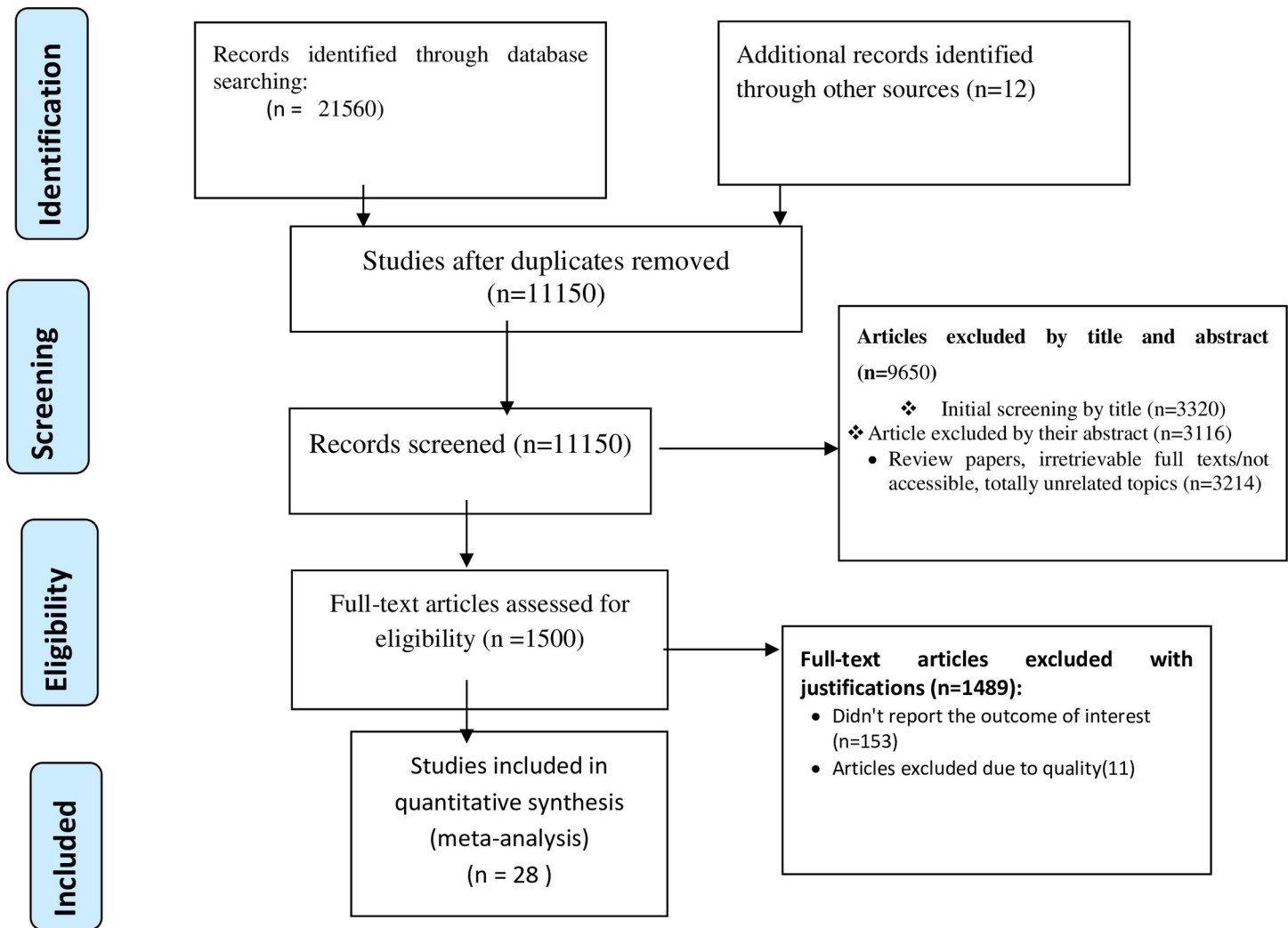

**Fig 1. PRISMA flow diagram showed the results of the search and reasons for exclusion.**

### Inclusion and exclusion criteria of studies included in the meta-analysis

Almost all included randomized control trials used similar eligibility criteria. They used the following inclusion criteria: 5 min or 10 min Apgar score, cord PH 7.1, base deficit, GA> = 36 weeks, BW> = 2500 g, and encephalopathy. On the other hand, Major congenital malformation, metabolic disorder, chromosomal abnormalities, congenital infection, persistent pulmonary hypertension, premature rupture of membranes, and >6 h of age used as exclusion criteria (Table 4).

### Meta-analysis

**The effects of therapeutic hypothermia on death among asphyxiated neonates with hypoxic-ischemic encephalopathy.** All of the studies (n = 28) reported the magnitude of mortality among cooled and non-cooled neonates with hypoxic-ischemic encephalopathy [20, 21, 28–50]. The authors calculated the relative risk of mortality in all included studies. The relative risk of mortality ranged from 0.00 l [20] up to 0.95 [21].

**Table 1. Distribution of studies on the effects of therapeutic hypothermia on death among asphyxiated neonates with hypoxic-ischemic encephalopathy.**

| Authors | Year | Country | Income | N Hypo: STD | Cooling method/ Device | Mortality HYPO: STD | Relative risk | Yes Total | Overall appraisal |
|---|---|---|---|---|---|---|---|---|---|
| Lin et al. [28] | 2006 | China | upper-middle | 32:30 | SHC: cooling caps | 2:2 | 0.94 | 12/13 | Included |
| Zhou et al. [29] | 2010 | China | upper-middle | 138:118 | SHC: cooling caps | 31:46 | 0.85 | 13/13 | Included |
| Akisu et al. [20] | 2013 | Turkey | upper-middle | 11:10 | SHC: cooling caps | 0:2 | 0 | 13/13 | Included |
| Robertson et al. [30] | 2008 | Uganda | Low | 21:15 | WBC: cooling blanket | 1:7 | 0.1 | 11/13 | Included |
| Thayyil et al. [31] | 2013 | India | low-middle | 17:16 | WBC: cooling blanket | 4:2 | 3.2 | 12/13 | Included |
| Bharadwaj et al. [32] | 2012 | India | low-middle | 62:62 | WHC: cold gel pack | 3:6 | 0.5 | 11/13 | Included |
| Bhat et al. [33] | 2006 | India | low-middle | 20:15 | WHC: cold gel pack | 3:5 | 0.45 | 13/13 | Included |
| Azzoparadi et al. [21] | 2009 | UK | High | 163:162 | WHC: cooling blanket | 42:44 | 0.86 | 12/13 | Included |
| Jacobs et al. [34] | 2011 | Australia | High | 91:78 | WHC: cold gel pack | 51:58 | 0.75 | 12/13 | Included |
| Shankaran et al. [35] | 2005 | USA | High | 102:103 | WHC: cooling blanket | 45:64 | 0.71 | 12/13 | Included |
| Simbruner et al. [36] | 2010 | Germany | High | 53:58 | WHC: cooling blanket | 27:48 | 0.62 | 11/13 | Included |
| Gluckman et al. [37] | 2005 | USA | High | 108:110 | SHC: cooling caps | 36:42 | 0.82 | 13/13 | Included |
| Zhou et al. [29] | 2010 | China | upper-middle | 100:94 | SHC: cooling caps | 31:46 | 0.65 | 12/13 | Included |
| Eicher et al. [38] | 2005 | USA | High | 32:33 | WHC: cooling blanket | 10:14 | 0.75 | 13/13 | Included |
| Battin et al. [39] | 2003 | New Zealand | High | 13:13 | SHC: cooling caps | 5:7 | 0.72 | 12/13 | Included |
| Shankaran et al. [40] | 2002 | USA | High | 9:10 | WHC: cooling blanket | 2:3 | 0.74 | 13/13 | Included |
| Joy et al. [41] | 2012 | India | low-middle | 58:58 | WHC: cold gel pack | 1:4 | 0.25 | 11/13 | Included |
| Maoulainine et al. [42] | 2017 | Morocco | Low | 19:19 | SHC; cooling caps | 3:7 | 0.43 | 13/13 | Included |
| Laptook et al. [43] | 2018 | USA | High | 83:85 | WBC: cooling blanket | 6:5 | 0.86 | 12/13 | Included |
| Gane et al. [44] | 2013 | India | low-middle | 53:50 | WBC: cold gel pack | 4:8 | 0.5 | 13/13 | Included |
| Selway et al [45] | 2010 | USA | High | 102:103 | WBC: cold gel pack | 24:38 | 0.65 | 12/13 | Included |
| Susan et al. [34] | 2011 | USA | High | 110:111 | WBC: cold gel pack | 55:67 | 0.77 | 13/13 | Included |
| Jose et al. [46] | 2018 | India | low-middle | 74:70 | WBC: cold gel pack | 18:28 | 0.6 | 12/13 | Included |
| Azzopardi et al. [47] | 2014 | UK | High | 145:135 | WHC: cooling blanket | 47:49 | 0.95 | 10/13 | Included |
| Shankaran et al. [48] | 2012 | USA | High | 58:43 | WBC: cold gel pack | 6:7 | 0.64 | 10/13 | Included |
| Battin et al. [49] | 2001 | New Zealand | High | 20:20 | SHC: cooling caps | 3:4 | 0.75 | 11/13 | Included |
| Gane et al. [44] | 2013 | India | low-middle | 60:60 | WBC: cold gel pack | 4:8 | 0.5 | 10/13 | Included |
| Namasivayam et al. [43] | 2017 | USA | High | 78:79 | WBC: cold gel pack | 19:22 | 0.85 | 11/13 | Included |

The fixed-effects model analysis from those studies revealed that the pooled relative risk of mortality was found to be 0.74 (95%CI; 0.67, 0.80; $I^2$ = 0.0%; p<0.996) (Fig 2).

**Subgroup analysis of the effects of therapeutic hypothermia on death among asphyxiated neonates with hypoxic-ischemic encephalopathy.** The subgroup analysis was done through stratifying by country income level, method of cooling, and device of cooling. Based on this, the pooled relative risk of mortality was 0.32, 0.5, 0.62, and 0.76 in the low, low middle, upper-middle, and high-income countries respectively (Fig 3 and Table 5). The relative risk of mortality was the same in selective head cooling and whole-body cooling method which was 0.74 (Fig 4 and Table 5).

Regarding the cooling device, the pooled relative risk of mortality is the same between the cooling cap and cooling blanket (0.74). However, it is slightly lower (0.73) in the cold gel pack (Fig 5 and Table 5).

**Table 2. Neonatal baseline characteristics of included studies on the effects of therapeutic hypothermia on death among asphyxiated neonates with hypoxic-ischemic encephalopathy.**

| SR no | Authors | Year | Mean BW (g) HYPO: STD | GA (weeks) HYPO: STD | Apgar score HYPO: STD | Moderate NE (n–%) HYPO: STD | Severe NE (n–%) HYPO: STD |
|---|---|---|---|---|---|---|---|
| 1. | Lin et al. [28] | 2006 | 3310: 3430 | 38.7: 39.1 | 3:3 | 16: 15 | 7: 6 |
| 2. | Zhou et al. [29] | 2010 | 3360: 3299 | NA: NA | NA: NA | 41: 41 | 38: 35 |
| 3. | Akisu et al. [20] | 2013 | 3410: 3270 | 39.3: 39.1 | 4.3: 4.1 | 7: 5 | 3:3 |
| 4. | Robertson et al. [30] | 2008 | 3300: 3200 | 38: 38 | 4.7: 5.2 | 10: 10 | 6:1 |
| 5. | Thayyil et al. [31] | 2013 | 2977: 2890 | 38: 38.9 | 4.3: 4.5 | 6: 5 | 2:2 |
| 6. | Bharadwaj et al. [32] | 2012 | 2967: 2899 | 39.8: 40 | NA: NA | 5.34: 5.26 | 55:7 |
| 7. | Bhat et al. [33] | 2006 | NA: NA | NA: NA | NA: NA | NA:NA | NA:NA |
| 8. | Azzopardi et al. [21] | 2009 | 3450: 3350 | 40.3: 40.1 | 4:4: 4 | 65: 57 | 98: 95 |
| 9. | Jacobs et al. [34] | 2011 | NA: NA | NA: NA | NA: NA | NA: NA | NA:NA |
| 10. | Shankaran et al. [35] | 2005 | 3385: 3370 | NA: NA | NA: NA | 69: 66 | 32: 40 |
| 11. | Simbruner et al. [36] | 2010 | 3300: 3300 | 39.2: 39.4 | 3.4: 3.4 | 24: 17 | 38: 46 |
| 12. | Gluckman et al. [37] | 2005 | 3399: 3504 | 38.9: 39.1 | NA: NA | NA: NA | NA:NA |
| 13. | Zhou et al. [29] | 2010 | 3360: 3299 | NA: NA | NA: NA | 41: 41 | 38: 35 |
| 14. | **Eicher et al. [38]** | 2005 | 3241: 3550 | 38.8: 39.1 | NA: NA | 5:5 | 25: 25 |
| 15. | Battin et al. [39] | 2003 | 3333: 3371 | 40: 39.8 | 5: 5 | NA: NA | NA:NA |
| 16. | Shankaran et al. [40] | 2002 | NA: NA | NA: NA | NA: NA | NA: NA | NA:NA |
| 17. | Joy et al. [41] | 2012 | 2840: 2910 | NA: NA | 3.81: 3.45 | 51:9 | 7:7 |
| 18. | Maoulainine et al. [42] | 2017 | 3336: 3300 | NA: NA | NA: NA | NA: NA | NA:NA |
| 19. | Laptook et al. [43] | 2018 | 3379: 3303 | 39:39 | 4.2: 4.3 | 73: 78 | 10:7 |
| 20. | Gane et al. [44] | 2013 | 2792: 2903 | NA: NA | 3: 3 | 44: 45 | 15: 16 |
| 21. | Selway et al. [45] | 2010 | NA: NA | NA: NA | NA: NA | NA: NA | NA: NA |
| 22. | Susan et al. [34] | 2011 | 3348: 3515 | 3939.2 | 3: 3 | 63: 54 | 30: 29 |
| 23. | Jose et al. [46] | 2018 | 3259: 3278 | NA: NA | NA: NA | 51: 39 | 22: 28 |
| 24. | Azzopardi et al. [47] | 2014 | 3467: 3351 | 40.3: 40.1 | NA: NA | 62: 57 | 83: 78 |
| 25. | Shankaran, et al. [48] | 2012 | 3172: 3555 | NA: NA | NA: NA | 28: 41 | 15: 16 |
| 26. | Battin, et al. [49] | 2001 | 3333: 3458 | 39.5: 39.4 | 5: 5 | NA: NA | NA:NA |
| 27. | Gane, et al. [44] | 2013 | 2792: 2903 | NA: NA | 3: 3 | 45: 44 | 15: 16 |
| 28. | Namasivayam et al. [43] | 2017 | NA: NA | NA: NA | NA: NA | NA: NA | NA:NA |

**Sensitivity analysis.** We employed a leave-one-out sensitivity analysis to identify the impact of the individual study on the pooled relative risk of mortality. The results of this sensitivity analysis showed that our findings were not dependent on a single study. Our pooled estimated relative risk varied between 0.72(0.65, 0.79) [36] and 0.75(0.68, 0.82) [21] after the deletion of a single study (S1 Fig).

**Publication bias.** We have also checked publication bias and a funnel plot showed symmetrical distribution (S2 Fig). Egger's regression test p-value was 0.156, which indicated the absence of publication bias (S3 Fig).

## Discussion

This systematic review and meta-analysis of RCTs was conducted to assess the effectiveness of therapeutic hypothermia/cooling therapy to reduce mortality of asphyxiated neonates with hypoxic-ischemic encephalopathy. Therapeutic hypothermia is found to be effective to reduce the risk of death in neonates with moderate to severe hypoxic-ischemic encephalopathy. In

**Table 3. Characteristics of excluded studies on the effects of therapeutic hypothermia on death among asphyxiated neonates with hypoxic-ischemic encephalopathy.**

| Excluded Studies | Country | Cooling method | Device | N | Comments | Reasons for exclusion |
|---|---|---|---|---|---|---|
| Horn [56] | South Africa | Selective head cooling | Frozen gel packs | 4 | Due to wide temperature fluctuations, the study stopped prematurely | Case series |
| Thomas [57] | India | Whole-body cooling | Frozen gel packs | 20 | The mean rectal temperature during cooling was 32.96 oC. | Case series |
| Horn [58] | South Africa | Selective head cooling | Selective head | 5 | A pilot study with frozen gel packs around the head | Case series |
| Rajhans [59] | India | Whole-body cooling | Blanketrol II | 5 | Only two babies completed cooling for 72 hours | Case series |
| Horn [60] | South Africa | Selective head cooling | Servo controlled Fan | 10 | Excessive shivering reported in the cooled infants. | Case series |
| Robertson [61] | Uganda | Whole-body cooling | Water bottles | 56 | Study protocol of a previously published cooling trial. | Protocol only |
| Thomas [57] | India | Whole-body cooling | Frozen gel packs | 14 | The adverse outcome was seen in 3 (2 deaths, 1 developmental delay) of the 14 infants (out of 20) | Case series |
| Li [62] | China | Whole-body cooling | Not described | 93 | Hypothermic induced within 10 hours, maintaining rectal temperature 33.5uC for 72 hours. | Study details unclear. |
| See [63] | Malaysia | Whole-body cooling | Ambient Temperature | 17 | Cooled by manipulating environmental temperature; report no neurological deficit in 14/15 stage 2 NE babies. | Case series |
| Horn [64] | South Africa | Selective Head Cooling | Frozen gel packs | 14 | Active rewarming using a radiant warmer | Case series |
| Tan [53] | Uganda | Whole-body cooling | Water bottles | 19 | One year follow up of previously recruited infants from a cooling trial. | Duplicate data |

addition, both selective head cooling and whole-body cooling methods are effective in reducing the mortality of infants with this condition.

The pooled relative risk of mortality among birth asphyxiated neonates who have got cooling therapy was found to be nearly 26% lower compared with those who haven't got cooling therapy. This result was similar with a systematic review and meta-analysis conducted in 2010 and 2013 [3, 15, 17, 51].

The above-mentioned similarity between our finding and others can be explained by different scientific assumption. Pathophysiologically, it is known that birth asphyxia leads to hypoxia and hypoxic ischemic insult. Initially, hypoxic ischemic (HI) insult results in primary energy failure which is characterized by decreased ATP production. This in turn leads to loss of integrity of the neuronal cell membrane, with calcium entry into the cell facilitated by activation of NMDA receptor and other excitotoxic neurotransmitters. At this stage, decreasing cerebral metabolism, antagonizing NMDA receptors and suppressing excitotoxic neurotransmitters are fundamental interventional strategies to be used to reverse brain damage/ treat HIE in asphyxiated neonates. In the absence of any intervention, secondary energy failure associated with moderate to severe HIE will occur after 6–48 hours' period of latency due to oxidative stress, inflammation, and ultimately leads to cell death. At this stage, interventional strategies targeted to reduce oxidative stress markers, inflammation and cell death are crucial to treat HIE in asphyxiated neonates [44, 52].

Another mechanism could be through reducing cerebral metabolism by inhibiting post depolarization release of many toxins. It can also reduce oxidative stress-induced DNA damage by reducing stress markers, attenuate excitatory brain damage, and suppress inflammation and programmed cell death (apoptosis) [44]. So, it is assumed that asphyxiated neonates who got cooling therapy will have reduced risk of mortality than neonates without cooling therapy.

**Table 4. Inclusion and exclusion criteria of included studies on the effects of therapeutic hypothermia on death among asphyxiated neonates with hypoxic-ischemic encephalopathy.**

| Authors | Inclusion criteria | Exclusion criteria |
|---|---|---|
| Lin ZL et al. [28] | 5 min Apgar,6 AND Cord pH,7.1 or base deficit .15 mmol/L AND encephalopathy | Major congenital abnormalities, persistent pulmonary<br>Hypertension |
| Zhou WH et al. [29] | 5 min Apgar,6 AND Cord pH,7 or base deficit #16 mmol/L AND need for resuscitation at 5 minutes of age | Major congenital abnormalities, maternal fever<br>.38uC, infection, rupture of membranes .18<br>hours or foul-smelling liquor, another encephalopathy |
| Akisu M et al. [20] | 5 min Apgar,6 AND Cord pH,7.1 or base deficit .10 mmol/L AND encephalopathy | Major congenital malformation, metabolic disorder, chromosomal abnormalities, congenital infection, transitory drug depression |
| Robertson et al. [30] | 5 min Apgar,6 AND encephalopathy (Thompson score .5) | Apnoea or cyanosis, absent cardiac output .10 min |
| Thayyil S et al. [31] | 5 min Apgar,6 AND encephalopathy (Thompson score .5) | Major congenital malformations, Imminent death at the time of randomization |
| Bharadwaj et al. [32] | 10 min Apgar,6 AND arterial pH#7 or base excess $12 meq AND encephalopathy | Major congenital abnormalities, no spontaneous respiration by 20 min, outborn babies |
| Bhat M et al. [33] | 10 minute Apgar,5 AND Cord pH,7 and or base deficit of .18 meq/L | Major congenital abnormalities, persistent pulmonary<br>Hypertension |
| Azzoparadi et al. [21] | GA> = 36 weeks with PHI, moderate to severe encephalopathy, and abnormal background on aEEG | Major congenital abnormalities or >6 h of age |
| Jacobs et al. [34] | GA > = 35 weeks with PHI and moderate or severe encephalopathy | Major congenital abnormalities, >6 h of age, BW <2 kg, overt bleeding, required >80% oxygen, death was imminent, or therapeutic hypothermia had commenced before assessment |
| Shankaran et al. [35] | GA> = 36 weeks with PHI, <6 h of age, and encephalopathy or seizures | Major congenital abnormalities, BW < = 1800 g, or >6 h of age |
| Simbruner et al. [36] | GA> = 36 weeks, PHI, encephalopathy, and abnormal EEG or aEEG findings | Major congenital malformations, >5.5 h of age, received anticonvulsant therapy, BW <1800 g, HC less than the third percentile for GA if BW and length are greater than the third percentile, imperforate anus, or gross hemorrhage |
| Gluckman et al. [37] | GA> = 36 weeks with PHI, moderate to severe encephalopathy, and abnormal background on an EEG | Major congenital abnormalities, >5.5 h of age, received prophylactic anticonvulsants, BW <1800 g, HC <2 SD for gestation if BW and length >−2 SD, or critically ill and unlikely to benefit from intensive care |
| Zhou et al. [29] | GA> = 37 weeks, BW> = 2500 g, PHI, and encephalopathy | Major congenital abnormalities, signs of infection, other causes of encephalopathy or severe anemia |
| Eicher DJ et al. [38] | > = 35 weeks gestation, > = 2000 gm birth weight, were <6 hours after birth and encephalopathy | Neonates with clinical sepsis, maternal chorioamnionitis, weight or head circumference less than 10th percentile for gestation age, or congenital abnormalities were excluded |
| Battin MR et al. [39] | 1) gestational age >37 weeks; 2) 5-minute Apgar score below 6 or cord/first arterial pH <7.1; and 3) encephalopathy | Major congenital abnormalities or those who presented to National Women's Hospital neonatal unit after 6 hours of age |
| Shankaran et al. [40] | All term infants who were >36 weeks' gestation and admitted to the neonatal intensive care unit at below 6 hours of age | 1) inability to perform random assignment by 6 hours of age, 2) chromosomal abnormality, 3) major congenital anomaly, 4) severe growth restriction (< = 1800 g birth weight), 5) infant unlikely to survive |
| Joy R et al. [41] | GA> = 37 weeks, BW> = 2500 g, PHI, and encephalopathy | Major congenital abnormalities, signs of infection, other causes of encephalopathy or severe anemia |
| Maoulainine et al. [42] | GA> = 36 weeks with PHI, <6 h of age, and encephalopathy or seizures | Major congenital abnormalities, BW < = 1800 g, or >6 h of age |
| Laptook AR et al. [43] | GA> = 36 weeks with PHI, moderate to severe encephalopathy, and abnormal background on an EEG | Major congenital abnormalities or >6 h of age |
| Gane B. D et al. [44] | > = 37 weeks with umbilical cord blood or arterial blood (within the first postnatal hour) PH < = 7 or base deficit > = 16 meq with evidence of encephalopathy | more than 6 h of age at the time of randomization, had major congenital abnormalities, did not establish spontaneous respiration by 20 min after birth |
| Selway L et al. [45] | 1) gestational age >37 weeks; 2) 5-minute Apgar score below 6 or cord/first arterial pH <7.1; and 3) encephalopathy | Major congenital abnormalities or those who presented to National Women's Hospital neonatal unit after 6 hours of age |
| Susan E. et al. [34] | 35 weeks' gestation or more at birth, could have hypothermia initiated within 6 hours of birth, had moderate or severe encephalopathy | hypothermia could not start within6 hours of birth if the birth weight was less than 2 kg if major congenital abnormalities were suspected |
| Jose S et al. [46] | moderate and severe encephalopathy within 6 hours after birth after an acute perinatal event | Major congenital abnormalities, signs of infection, other causes of encephalopathy or severe anemia |

*(Continued)*

**Table 4.** (Continued)

| Authors | Inclusion criteria | Exclusion criteria |
|---|---|---|
| Azzopardi M.D et al.[47] | GA> = 37 weeks, BW> = 2500 g, PHI, and encephalopathy | Major congenital abnormalities, signs of infection, other causes of encephalopathy or severe anemia |
| Shankaran, MD et al. [48] | GA> = 36 weeks with PHI, <6 h of age, and encephalopathy or seizures | Major congenital abnormalities, BW < = 1800 g, or >6 h of age |
| Battin, M. R et al. [49] | GA> = 36 weeks with PHI, <6 h of age, and encephalopathy or seizures | Major congenital abnormalities, BW < = 1800 g, or >6 h of age |
| Gane, B. D et al. [44] | GA> = 37 weeks, BW> = 2500 g, PHI, and encephalopathy | Major congenital abnormalities, signs of infection, other causes of encephalopathy or severe anemia |
| Namasivayam A et al. [43] | GA> = 36 weeks with PHI, moderate to severe encephalopathy, and abnormal background on aEEG | Major congenital abnormalities or >6 h of age |

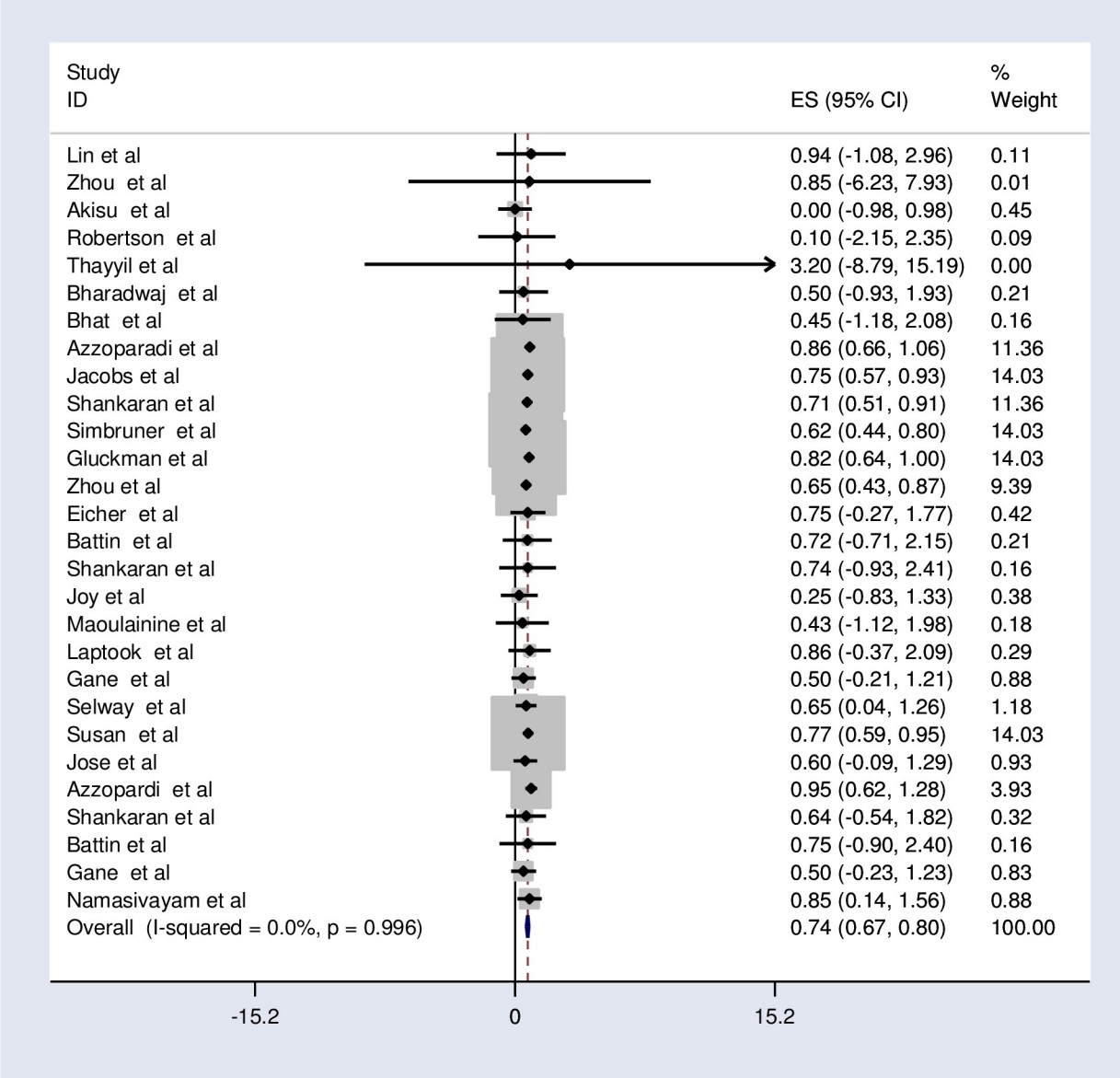

**Fig 2. Forest plot showing the effects of therapeutic hypothermia on death among asphyxiated neonates with hypoxic-ischemic encephalopathy.**

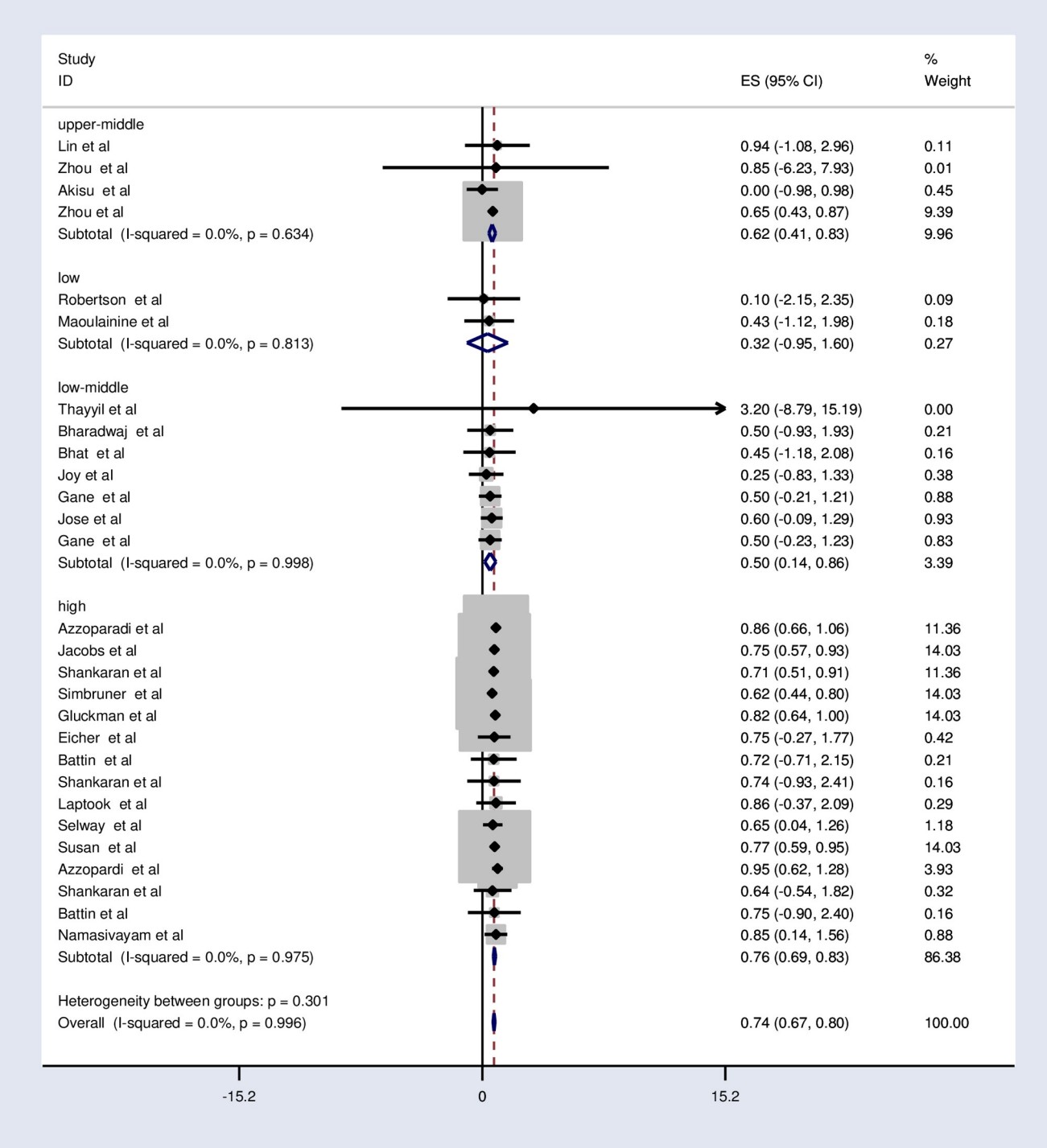

**Fig 3. Subgroup analysis by the country level of income on the effects of therapeutic hypothermia on death among asphyxiated neonates with hypoxic-ischemic encephalopathy.**

The subgroup analysis by income in our study found that cooling therapy can reduce mortality of asphyxiated neonates in low and middle income countries better than in high income countries. In contrast to this result, a systematic review and meta-analysis conducted in low

**Table 5. Subgroup analysis of the effects of therapeutic hypothermia on death among asphyxiated neonates with hypoxic-ischemic encephalopathy.**

| Variables | Characteristics | Pooled prevalence (95% CI) | $I^2$(P-value) |
|---|---|---|---|
| Country income level | High | 0.76 (0.69, 0.83) | 0.0% (0.975) |
| | Upper middle | 0.62 (0.41, 0.83) | 0.0% (0.634) |
| | Low-middle | 0.50 (0.14, 0.86) | 0.0% (0.998) |
| | Low | 0.32 (-0.95, 1.60) | 0.0% (0.813) |
| Methods of cooling | Selective head cooling | 0.74 (0.60, 0.87) | 0.0% (0.798) |
| | Whole body cooling | 0.74 (0.66, 0.81) | 0.0% (0.998) |
| Device of cooling | Cooling caps | 0.74 (0.60, 0.87) | 0.0% (0.798) |
| | Cooling blanket | 0.74 (0.64, 0.85) | 0.0% (0.721) |
| | Cold gel pack | 0.73 (0.62, 0.84) | 0.0% (0.993) |

and middle income (LMI) countries found no significant reduction of neonatal mortality with cooling therapy in those countries. But, it had failed to exclude clinically important benefits/ harms of cooling therapy due to wide CI. Rather, it had explained as the apparent lack of treatment effect might be due to the heterogeneity and poor quality of the included studies, inefficiency of the low technology cooling devices, lack of optimal neonatal intensive care [53].

In line with results of this meta-analysis, literatures had revealed that cooling therapy can reduce mortality of asphyxiated neonates in LMI countries [15, 53]. However, safety and affordability of cooling therapy in those countries was under question [52]. Since this meta-analysis had not explored safety and affordability issues of cooling therapy, authors had failed to strongly praise direct application of cooling therapy in LMI countries. According to ILCOR (2020) cooling treatment should be considered when neonatal care facilities fulfill infrastructures and adequate resources to offer intravenous therapy, respiratory support, pulseoximetry, antibiotics, anticonvulsants, and pathology testing [19].

Concerning cooling methods, the relative risk of mortality among asphyxiated neonates who got selective head cooling therapy or whole-body cooling therapy was found to be the same. A meta-analysis conducted in 2012 had revealed a slightly reduced risk of mortality in neonates who got whole body cooling therapy than neonates with selective head cooling therapy [51]. Literature indicated that, even if both whole-body and selective head cooling are effective methods to provide cooling therapy and have comparable outcomes, whole-body cooling is more commonly used due to the ease of administration. Selective head cooling is more problematic (vulnerable for high temperature fluctuations and hyperthermia during rewarming) which makes it difficult for clinical application. Whole body cooling provides systemic effect with cooling of almost all parts of the brain, while selective head cooling cools only cortical part of the brain [28, 54, 55]. Due to these reasons, authors of this meta-analysis believe that whole body cooling is more applicable and effective than selective head cooling; but it needs further research and explanation.

Regarding cooling devices, the pooled relative risk of mortality was found to be the same between cooling cap and cooling blanket. However, it was slightly lower in cold gel pack. Literatures comparing cooling cap, cooling blanket and cold gel pack in terms of effectiveness were not found. So, authors of this meta-analysis suggest the need for further researches on this issue.

## Strength and limitations

This meta-analysis has several strengths. One is absence of heterogeneity among included randomized control trials in all pooling analysis. Besides, included randomized control trials have

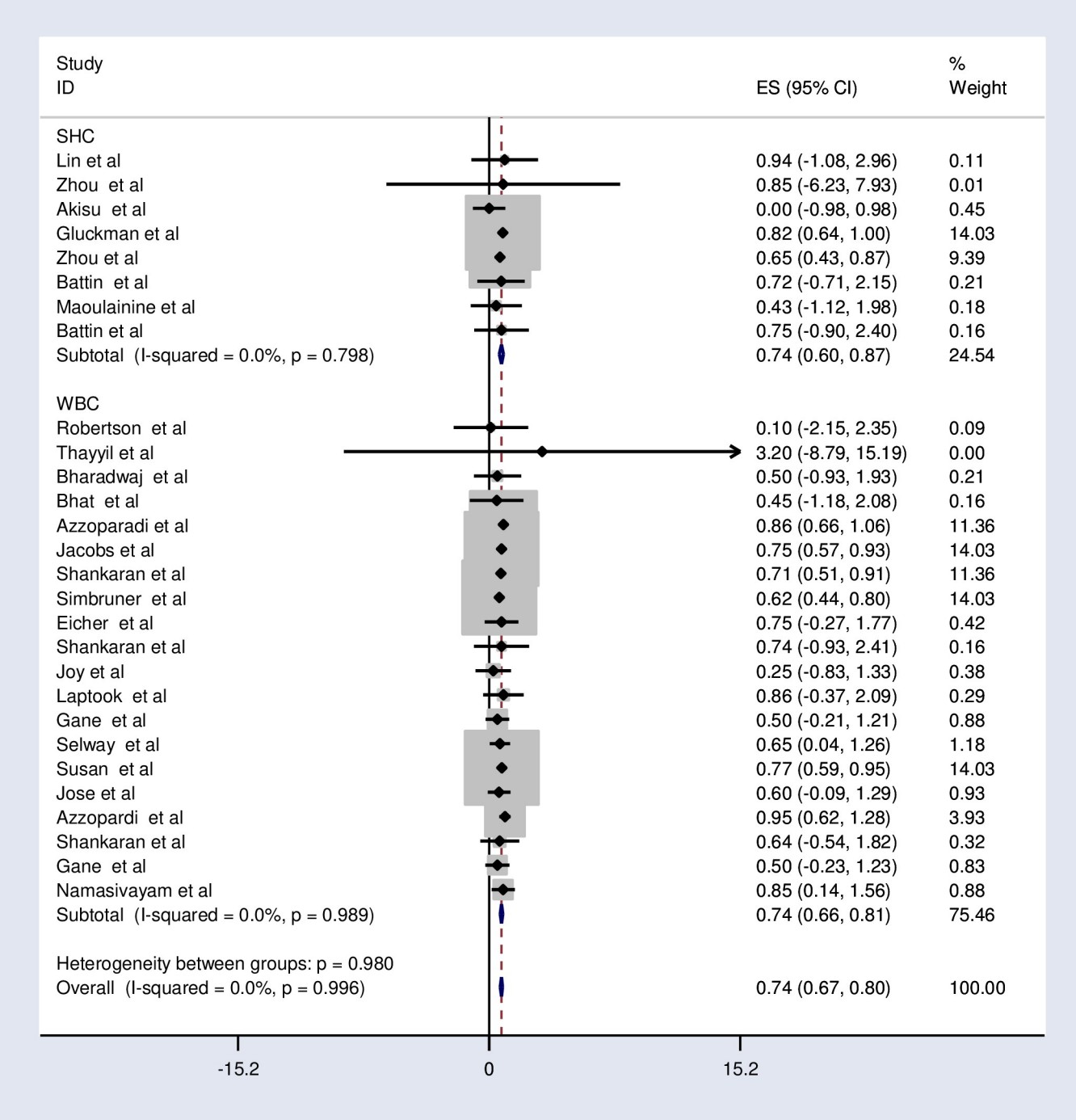

**Fig 4. Subgroup analysis by the method of cooling effects of therapeutic hypothermia on death among asphyxiated neonates with hypoxic-ischemic encephalopathy.**

high quality as assessed by JBI quality appraisal checklist for randomized control trials. This study also has certain limitations. First, this systematic review and meta-analysis only assessed the impact of therapeutic hypothermia on mortality. It lacks data on the impact of reducing

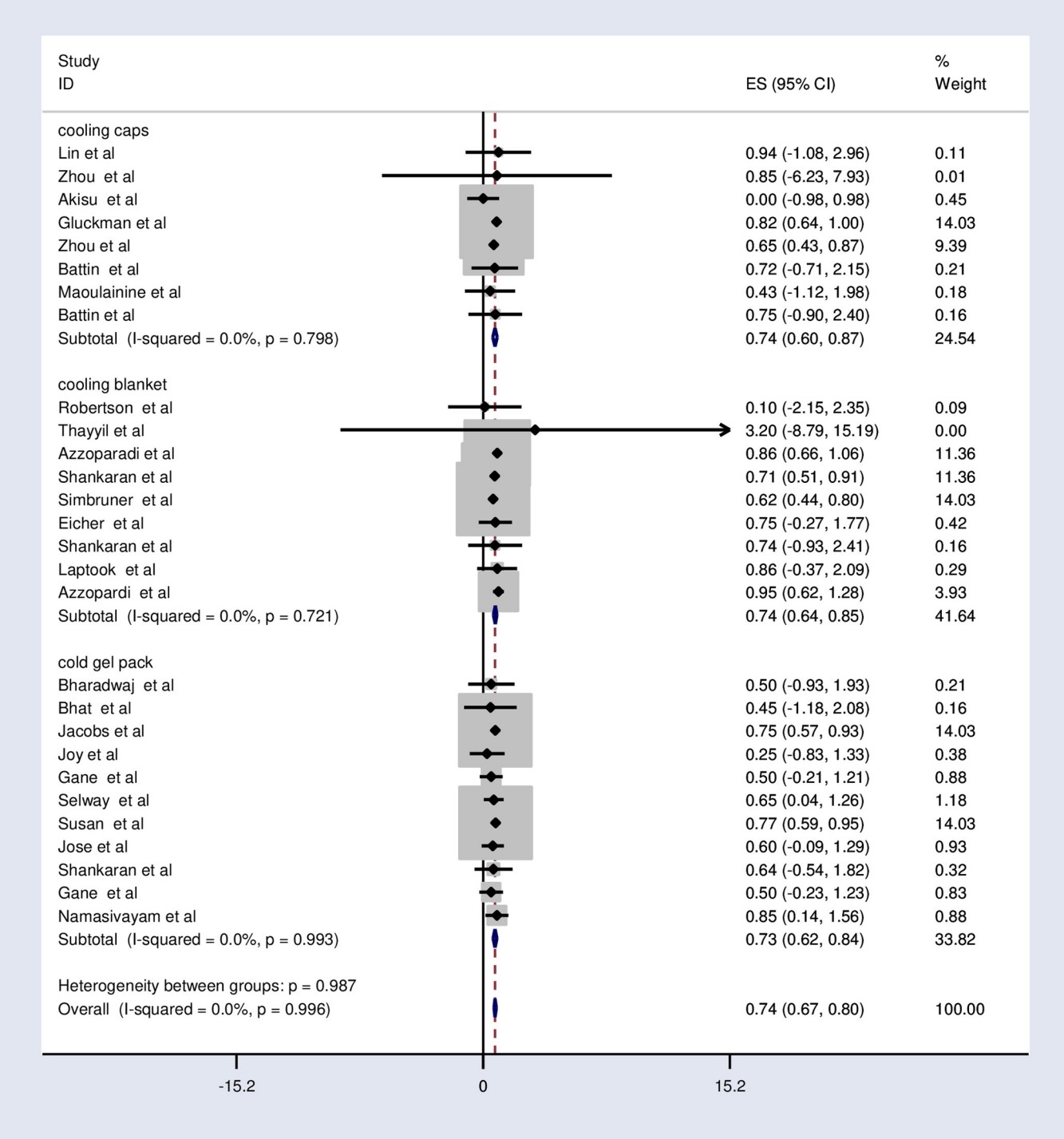

**Fig 5. Subgroup analysis by the device of cooling effects of therapeutic hypothermia on death among asphyxiated neonates with hypoxic-ischemic encephalopathy.**

disabilities and chronic complications among survived infants. Moreover, safety and affordability issues of applying therapeutic hypothermia in LMIC is not addressed here and needs further investigation.

## Conclusion

Therapeutic hypothermia reduces the risk of death in neonates with moderate to severe hypoxic-ischemic encephalopathy. Both selective head cooling and whole-body cooling method are equally effective in reducing the mortality of infants with this condition. Cold gel pack was slightly better than the cooling cap and cooling blanket in reducing mortality. Cooling therapy can be applied by using low-cost servo-controlled cooling devices/ low technology devices like ice pack, cold gel pack, cooling cap, cooling fans, cooling blanket, water bottles and others. Therefore, health professionals should consider offering therapeutic hypothermia as part of routine clinical care to newborns with hypoxic-ischemic encephalopathy especially in low- and middle-income countries after exploring safety issues with fulfillment of neonatal care facilities infrastructures and adequate resources to offer multi care like intravenous therapy, respiratory support, pulseoximetry, antibiotics, anticonvulsants, and pathology testing. Additional randomized control trial targeting safety, affordability and effective method of cooling and cooling devices to be applied in LMI countries should also be investigated.

## Supporting information

**S1 Checklist. PRISMA checklist.**
(DOCX)

**S1 Fig.** Sensitivity analysis the effects of therapeutic hypothermia on death among asphyxiated neonates with hypoxic ischemic encephalopathy.
(DOCX)

**S2 Fig. Funnel plot showing publication bias on the effects of therapeutic hypothermia on death among asphyxiated neonates with hypoxic ischemic encephalopathy.**
(DOCX)

**S3 Fig. Eggers test showing publication bias on the effects of therapeutic hypothermia on death among asphyxiated neonates with hypoxic ischemic encephalopathy.**
(DOCX)

**S1 Table. Search strategy used for one of the databases.**
(DOCX)

**S2 Table. Quality appraisal result of included studies; using Joanna Briggs Institute (JBI) quality appraisal checklist.**
(DOCX)

**S1 Synopsis.**
(DOCX)

## Author Contributions

**Conceptualization:** Biruk Beletew Abate, Melaku Bimerew, Bereket Gebremichael, Ayelign Mengesha Kassie, MesfinWudu Kassaw, Teshome Gebremeskel, Wubet Alebachew Bayih.

**Data curation:** Biruk Beletew Abate, Melaku Bimerew, Ayelign Mengesha Kassie, Mesfin-Wudu Kassaw, Teshome Gebremeskel, Wubet Alebachew Bayih.

**Formal analysis:** Biruk Beletew Abate, Melaku Bimerew, Ayelign Mengesha Kassie, Teshome Gebremeskel, Wubet Alebachew Bayih.

**Funding acquisition:** Biruk Beletew Abate, Melaku Bimerew, Teshome Gebremeskel.

**Investigation:** Biruk Beletew Abate, MesfinWudu Kassaw, Teshome Gebremeskel, Wubet Alebachew Bayih.

**Methodology:** Biruk Beletew Abate, Wubet Alebachew Bayih.

**Project administration:** Biruk Beletew Abate, Wubet Alebachew Bayih.

**Resources:** Biruk Beletew Abate, Melaku Bimerew, Wubet Alebachew Bayih.

**Software:** Biruk Beletew Abate, MesfinWudu Kassaw, Wubet Alebachew Bayih.

**Supervision:** Biruk Beletew Abate, Bereket Gebremichael, Ayelign Mengesha Kassie, Mesfin-Wudu Kassaw, Wubet Alebachew Bayih.

**Validation:** Biruk Beletew Abate, Bereket Gebremichael, Ayelign Mengesha Kassie, Wubet Alebachew Bayih.

**Visualization:** Biruk Beletew Abate, Bereket Gebremichael, MesfinWudu Kassaw, Wubet Alebachew Bayih.

**Writing – original draft:** Melaku Bimerew, MesfinWudu Kassaw, Teshome Gebremeskel, Wubet Alebachew Bayih.

**Writing – review & editing:** Biruk Beletew Abate, Melaku Bimerew, Bereket Gebremichael, Ayelign Mengesha Kassie, Teshome Gebremeskel, Wubet Alebachew Bayih.

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
