## [Decision Letter · Decision Letter 0]

10 Sep 2020

PONE-D-20-24388

Effects of therapeutic hypothermia on death among asphyxiated neonates with hypoxic-ischemic encephalopathy: a systematic review and meta-analysis of randomized control trials

PLOS ONE

Dear Dr. Biruk Beletew Abate,

Thank you for submitting your manuscript to PLOS ONE. After careful consideration, we feel that it has merit but does not fully meet PLOS ONE’s publication criteria as it currently stands. Therefore, we invite you to submit a revised version of the manuscript that addresses the points raised during the review process.

We look forward to receiving your revised manuscript.

Kind regards,

Georg M. Schmölzer

Academic Editor

PLOS ONE

Journal Requirements:

3. Please include a caption for figure 4.

Reviewers' comments:

Reviewer's Responses to Questions

**Comments to the Author**

1. Is the manuscript technically sound, and do the data support the conclusions?

Reviewer #1: No

2. Has the statistical analysis been performed appropriately and rigorously? 

Reviewer #1: I Don't Know

3. Have the authors made all data underlying the findings in their manuscript fully available?

Reviewer #1: Yes

4. Is the manuscript presented in an intelligible fashion and written in standard English?

Reviewer #1: Yes

5. Review Comments to the Author

Reviewer #1: This meta-analysis and systematic review by Biruk Beletew et al regarding effect of hypothermia on mortality in HIE newborn babies is important and timely. The authors basically confirm previous meta-analyses demonstrating a 25% reduction in mortality after cooling. However, the authors conclude that cooling is most efficient in low income countries. If this is correct, it is important because until recently there has been concerns regarding cooling in such areas due to higher mortality. Further, the authors find similar effects using head cooling and total body cooling.

I have some comments and concerns:

1.Introduction first sentence: Perhaps it is more appropriate to write: intrapartum and neonatal asphyxia. A modern term of birth asphyxia is however adverse intrapartum events

2. In the introduction the authors should mention that a powerful mean to reduce HIE is to avoid hyperoxia. See for instance:

Kapadia VS, Chalak LF, DuPont TL, Rollins NK, Brion LP, Wyckoff MH. Perinatal asphyxia with hyperoxemia within the first hour of life is associated with moderate to severe hypoxic-ischemic encephalopathy. J Pediatr. 2013;163(4):949-954.

Saugstad OD. The oxygen paradox in the newborn: keep oxygen at normal levels. J Pediatr. 2013;163(4):934-935.

and reducing death:

Saugstad OD, Ramji S, Soll RF, Vento M. Resuscitation of newborn infants with 21% or 100% oxygen: an updated systematic review and meta-analysis. Neonatology. 2008;94(3):176-182.

3. It would be useful to separate data in studies in which the newborns were resuscitated with 100% O2 (approx before 2010) and with air.

4. I have problems with some of the results. The authors find in general a reduced relative risk of death (RR 0.74). That is a 26 % reduction. how can this be turned to a 75% reduced risk of death in the discussion part? See: "The pooled relative risk of mortality among birth asphyxiated neonates who have got cooling therapy was found to be nearly 75% lower compared with those who haven’t got cooling therapy." Please explain

5. Regarding the results from low income countries two studies seem to be included. One of these is naturally Robertson NJ et al from Uganda 2008. In this study there was an approiximate 6 fold increased risk of death in the cooled babies compared to non cooled. (There was also more seizures in the cooled group). The Robertson study is small, still, based on this how is it possible to get a RR of 0.32 for death in low income countries ? The Robertson study has been very important for recommendations of being cautious using cooling in low income areas.

Conclusion

This study has many merits. It is well performed and well written. It deals with a highly important topic. My main objection is that I dont understand all the results. I look forward to a clarification by the authors.

6. PLOS authors have the option to publish the peer review history of their article (what does this mean?). If published, this will include your full peer review and any attached files.

Reviewer #1: **Yes: **Ola Didrik Saugstad

---

## [Author Response · Author response to Decision Letter 0]

20 Oct 2020

Date: 9/24/2020

To: "PLOS ONE" plosone@plos.org

From: "Biruk Beletew Abate" birukkelemb@plos.com

Subject: Point to point response / 'Response to Reviewers'.

PONE-D-20-24388

Effects of therapeutic hypothermia on death among asphyxiated neonates with hypoxic-ischemic encephalopathy: a systematic review and meta-analysis of randomized control trials

PLOS ONE

To Editor 

Dear Academic Editor (Georg M. Schmölzer) we have no word to explain our deepest thanks for your constructive comments and helping us throughout the process. Since we have agreed with all points you raised we believe we have carefully amended the paper as per your point of view. We have presented the point to point response below. Thank you again since you have contributed much for our better paper.

 Editor comment 1. Please ensure that your manuscript meets PLOS ONE's style requirements, including those for file naming. The PLOS ONE style templates can be found at

Authors’ response: Thanks for supplying us the PLOS ONE style templates and we make sure the manuscript is as per these templates. 

 Editor comment 2. Please include your tables as part of your main manuscript and remove the individual files. Please note that supplementary tables (should remain/ be uploaded) as separate "supporting information" files.

Authors’ response: We have included tables and figures as part of the manuscript and we have uploaded the supplementary tables and figures as separate "supporting information" files as per your recommendation.

Editor comment 3. Please include a caption for figure 4.

Authors’ response: We have included as per your comment

 Editor comment 4. Please include captions for your Supporting Information files at the end of your manuscript, and update any in-text citations to match accordingly. Please see our Supporting Information guidelines for more information: http://journals.plos.org/plosone/s/supporting-information.

Authors’ response: we have included captions for your Supporting Information files at the end of the manuscript

 To Reviewer #1: 

Dear Ola Didrik Saugstad (Reviewer) we would like to forward our deep-seated gratitude for your interesting and valuable comments. We all really appreciate your potential and optimism while you give such constructive and in-depth comments. Since we have agreed with all of your points raised we believe we have amended the manuscript as per your comments. We would like to thank you again since you are contributing for our better paper by giving such comments which is important to improve the quality of this paper. Below we have written the point to point response to issues you raised.

Reviewer comment: This meta-analysis and systematic review by Biruk Beletew et al regarding effect of hypothermia on mortality in HIE newborn babies is important and timely. The authors basically confirm previous meta-analyses demonstrating a 25% reduction in mortality after cooling. However, the authors conclude that cooling is most efficient in low income countries. If this is correct, it is important because until recently there has been concerns regarding cooling in such areas due to higher mortality. Further, the authors find similar effects using head cooling and total body cooling.

I have some comments and concerns:

Reviewer comment: 1.Introduction first sentence: Perhaps it is more appropriate to write: intrapartum and neonatal asphyxia. A modern term of birth asphyxia is however adverse intrapartum events

Authors’ response: we amended the sentence as per your comment: Introduction; page 3 line 50 and 51.

Reviewer comment: 2. In the introduction the authors should mention that a powerful mean to reduce HIE is to avoid hyperoxia. See for instance:

Kapadia VS, Chalak LF, DuPont TL, Rollins NK, Brion LP, Wyckoff MH. Perinatal asphyxia with hyperoxemia within the first hour of life is associated with moderate to severe hypoxic-ischemic encephalopathy. J Pediatr. 2013;163(4):949-954.

Saugstad OD. The oxygen paradox in the newborn: keep oxygen at normal levels. J Pediatr. 2013;163(4):934-935.

and reducing death:

Saugstad OD, Ramji S, Soll RF, Vento M. Resuscitation of newborn infants with 21% or 100% oxygen: an updated systematic review and meta-analysis. Neonatology. 2008;94(3):176-182.

Authors’ response: We really thank you for providing us such very important papers related to our manuscript. We have downloaded all the above references, used and cited in the introduction part of the manuscript as per your recommendation.

Reviewer comment: 3. It would be useful to separate data in studies in which the newborns were resuscitated with 100% O2 (approx before 2010) and with air.

Authors’ response: we included those studies which have similar resuscitation status which is the standard treatment including 21 % O2. That is why we didn’t conduct subgroup analysis by level of resuscitation separately.

Reviewer comment: 4. I have problems with some of the results. The authors find in general a reduced relative risk of death (RR 0.74). That is a 26 % reduction. how can this be turned to a 75% reduced risk of death in the discussion part? See: "The pooled relative risk of mortality among birth asphyxiated neonates who have got cooling therapy was found to be nearly 75% lower compared with those who haven’t got cooling therapy." Please explain

Authors’ response: It was by mistake we said 75%. Now we have amended it to 26%. Sorry for our mistake and thanks since you remembered us.

Reviewer comment: 5. Regarding the results from low income countries two studies seem to be included. One of these is naturally Robertson NJ et al from Uganda 2008. In this study there was an approiximate 6 fold increased risk of death in the cooled babies compared to non cooled. (There was also more seizures in the cooled group). The Robertson study is small, still, based on this how is it possible to get a RR of 0.32 for death in low income countries ? The Robertson study has been very important for recommendations of being cautious using cooling in low income areas.

Authors’ response: Regarding the Robertson study; the authors taken small number of studies which was 21:15 hypothermic and standard treatment respectively. As a result during pooling we are doing a weighted mean , that means studies which have large sample size will have high contribution on the pooled effect while studies like Robertson will have small contribution. While the other study, Maullanie et al, in low income country have a larger sample size and the RR is 0.43. That is why RR of 0.32 for death in low income which is near to 0.43. 

Reviewer comment: Conclusion

This study has many merits. It is well performed and well written. It deals with a highly important topic. My main objection is that I don’t understand all the results. I look forward to a clarification by the authors.

Authors’ response: we really thank you for the above constructive comments. Regarding the results; since, this systematic review and meta-analysis aimed to assess the effect of therapeutic hypothermia on death among asphyxiated neonates with hypoxic-ischemic encephalopathy. 

As a result we extracted the author's name, year of publication, country, method of cooling, the severity of encephalopathy, the sample size in the hypothermic, and non-hypothermic groups, and the number of deaths in the intervention and control groups. The primary outcome was the relative risk (RR) of death, which was calculated as the proportion of death among hypothermic over the proportion of death among non-hypothermic patients. We calculated the RR for each included studies. Finally we pooled the estimates RR of death. The pooled relative risk of mortality after implementation of therapeutic hypothermia was found to be 0.74 (95%CI; 0.67, 0.80; I2=0.0%; p<0.996). That means a 26% reduction in mortality is revealed among cooled (hypothermic) groups compared to non-hypothermic groups. The relative risk of mortality was the same in selective head cooling and whole-body cooling method which was 0.74. Regarding the cooling device, the pooled relative risk of mortality is the same between the cooling cap and cooling blanket (0.74).

---

## [Decision Letter · Decision Letter 1]

19 Jan 2021

PONE-D-20-24388R1

Effects of therapeutic hypothermia on death among asphyxiated neonates with hypoxic-ischemic encephalopathy: a systematic review and meta-analysis of randomized control trials

PLOS ONE

Dear Dr. Biruk Beletew Abate,

Thank you for submitting your manuscript to PLOS ONE. After careful consideration, we feel that it has merit but does not fully meet PLOS ONE’s publication criteria as it currently stands. Therefore, we invite you to submit a revised version of the manuscript that addresses the points raised during the review process.

We look forward to receiving your revised manuscript.

Kind regards,

Georg M. Schmölzer

Academic Editor

PLOS ONE

Reviewers' comments:

Reviewer's Responses to Questions

**Comments to the Author**

1. If the authors have adequately addressed your comments raised in a previous round of review and you feel that this manuscript is now acceptable for publication, you may indicate that here to bypass the “Comments to the Author” section, enter your conflict of interest statement in the “Confidential to Editor” section, and submit your "Accept" recommendation.

Reviewer #1: All comments have been addressed

2. Is the manuscript technically sound, and do the data support the conclusions?

Reviewer #1: Partly

3. Has the statistical analysis been performed appropriately and rigorously? 

Reviewer #1: I Don't Know

4. Have the authors made all data underlying the findings in their manuscript fully available?

Reviewer #1: Yes

5. Is the manuscript presented in an intelligible fashion and written in standard English?

Reviewer #1: No

6. Review Comments to the Author

Reviewer #1: This manuscript has been much improved and I want to congratulate the authors. I stlll have some comments

1. I am still concerned regarding the results of low income countries. Only two studies are available and listed, Robertson et al and Maoulainine et al. Robertson found higher mortality with hypothermia while Maoulainine et al found the opposite. Could the authors explain to me that in the table Robertson et al has an ES (is this relative risk?) of 0.10 while Maoulainine et al show 0.43. The fact is that Robertson's results are far above 1 although not significant (5. 0, 95 %CI 0,7-37). Maoulainine et al's study from Morrocco includes 38 babies (19 in each group) while Robetson et al include 36 babies , of these 21 treated with cooling and 15 received standard care. The size of these two studies therefore are quite similar.

2. I suggest the authors also refer to the most recent guidelines from ILCOR (2020) regarding hypothermia treatment cooling to only be considered in neonatal care facilities with the capabilities for “multidisciplinary care and availability of adequate resources to offer intravenous therapy, respiratory support, pulseoximetry, antibiotics, anticonvulsants, and pathology testing”

2. Some language issues

line 65: one of the strategies

line 113. What is "snowball search"?

i

line 152: Instead of "the mistyping" perhaps better is "any mistyping"

Line 186: methods head cooling. Remove "head"

Line 198, comma lacking, base deficit

Line: 236 ischemia not eschemia

line 238: methods

278: literature not literatures

Conclusion: A well performed study, and now much improved. Please check the calculations mentioned above and change conclusions accordingly.

7. PLOS authors have the option to publish the peer review history of their article (what does this mean?). If published, this will include your full peer review and any attached files.

Reviewer #1: **Yes: **Ola Didrik Saugstad

---

## [Author Response · Author response to Decision Letter 1]

2 Feb 2021

PONE-D-20-24388R1

Effects of therapeutic hypothermia on death among asphyxiated neonates with hypoxic-ischemic encephalopathy: a systematic review and meta-analysis of randomized control trials

PLOS ONE

Date: 9/24/2020

To: "PLOS ONE" plosone@plos.org

From: "Biruk Beletew Abate" birukkelemb@plos.com

Subject: Point to point response / 'Response to Reviewers'.

PONE-D-20-24388R1

Effects of therapeutic hypothermia on death among asphyxiated neonates with hypoxic-ischemic encephalopathy: a systematic review and meta-analysis of randomized control trials

PLOS ONE

To Editor 

Dear Academic Editor (Georg M. Schmölzer) we have no word to explain our deepest thanks for your constructive comments and helping us throughout the process. We have added one Author who fulfill the authorship criteria of the journal whom the corresponding author forgot during submission of the manuscript. For this we ask apologize. Thank you again since you have contributed much for our better paper.

To Reviewer #1: 

Dear Ola Didrik Saugstad (Reviewer) we would like to forward our deep-seated gratitude for your interesting and valuable comments. We all really appreciate your potential and optimism while you give such constructive and in-depth comments. Since we have agreed with all of your points raised we believe we have amended the manuscript as per your comments. We would like to thank you again since you contributed much the quality of this paper. Below we have written the point to point response to indicate we addressed the issues you raised.

Reviewer comment:: This manuscript has been much improved and I want to congratulate the authors. I still have some comments

Authors’ response: thanks since you recognized our effort, we have again amended the manuscript considering all your comments.

Reviewer comment: 1. I am still concerned regarding the results of low income countries. Only two studies are available and listed, Robertson et al and Maoulainine et al. Robertson found higher mortality with hypothermia while Maoulainine et al found the opposite. Could the authors explain to me that in the table Robertson et al has an ES (is this relative risk?) of 0.10 while Maoulainine et al show 0.43. The fact is that Robertson's results are far above 1 although not significant (5. 0, 95 %CI 0,7-37). Maoulainine et al's study from Morrocco includes 38 babies (19 in each group) while Robetson et al include 36 babies , of these 21 treated with cooling and 15 received standard care. The size of these two studies therefore are quite similar.

Authors’ response: as you explained from low income countries only two studies were included because these are the only studies which included the outcome of interest and fulfill the study’s inclusion criteria. However, you are correct this is the limitation of this study. Regarding the estimate in the two studies, you are again correct the Robertson's results not significant (5. 0, 95 % CI 0,7-37). 

For purpose of clarity, in Robertson's results the proportion of death among neonates with therapeutic hypothermia and standard treatment were 1/21and 7/15 respectively. Then the relative risk of death among neonates with cooling therapeutic becomes 0.10(Robertson NJ, Nakakeeto M, Hagmann C, Cowan FM, Acolet D, Iwata O, et al. Therapeutic hypothermia for birth asphyxia in low-resource settings: a pilot randomised controlled trial. The Lancet. 2008;372(9641):801-3). Meanwhile, Maoulainine et al's study from Morrocco includes 38 babies (19 in each group) 3:7 the proportion of death among neonates with theraputhic hypothermia and standard treatment were 3/19 and 7/19 respectively. Then the relative risk of death among neonates with cooling therapeutic becomes 0.43 (Maoulainine F, Elbaz M, Elfaiq S, Boufrioua G, Elalouani F, Barkane M, et al. Therapeutic hypothermia in asphyxiated neonates: Experience from neonatal intensive care unit of University Hospital of Marrakech. International journal of pediatrics. 2017;2017. The measure of the estimate was RR. Your comment is correct and we have tried to amend the manuscript as per your comment. Thanks for your detail noticing of the manuscript.

Reviewer comment: 2. I suggest the authors also refer to the most recent guidelines from ILCOR (2020) regarding hypothermia treatment cooling to only be considered in neonatal care facilities with the capabilities for “multidisciplinary care and availability of adequate resources to offer intravenous therapy, respiratory support, pulseoximetry, antibiotics, anticonvulsants, and pathology testing”.

Authors’ response: thanks for supplying us with such a latest source which we have included the idea the manuscript (Back ground; page 4, Line 92-95) and in Discussion ; page 13, Line 280-283).

2. Some language issues

Reviewer comment: line 65: one of the strategies

Authors’ response: amended; line 66

Reviewer comment: line 113. What is "snowball search"?

Authors’ response: "snowball search" is to refer the following process. When we get an article with the outcome of interest of the meta-analysis we then look at all references of that article and if we get any cross-referenced article which include our outcome of interest we will download the PDF and again we look at its references this process continues until we didn’t get any new article. We tried to indicate this when we say snowball. 

Reviewer comment: line 152: Instead of "the mistyping" perhaps better is "any mistyping"

Authors’ response: amended; line 158

Reviewer comment: Line 186: methods head cooling. Remove "head"

Authors’ response: amended; line 192

Reviewer comment: Line 198, comma lacking, base deficit

Authors’ response: amended; line 204

Reviewer comment: Line: 236 ischemia not eschemia

Authors’ response: amended; 238

Reviewer comment: line 238: methods

Authors’ response: amended; 240

Reviewer comment: 278: literature not literatures

Authors’ response: amended; 283

Reviewer comment: Conclusion: A well performed study, and now much improved. Please check the calculations mentioned above and change conclusions accordingly.

Authors’ response: we have amended the conclusion considering your comments in other parts of the manuscript. Line 308-320.

Thanks in advance

---

## [Editor Report · Decision Letter 2]

4 Feb 2021

Effects of therapeutic hypothermia on death among asphyxiated neonates with hypoxic-ischemic encephalopathy: a systematic review and meta-analysis of randomized control trials

PONE-D-20-24388R2

Dear Dr. Biruk Beletew Abate,

We’re pleased to inform you that your manuscript has been judged scientifically suitable for publication and will be formally accepted for publication once it meets all outstanding technical requirements.

Kind regards,

Georg M. Schmölzer

Academic Editor

PLOS ONE
---

## [Editor Report · Acceptance letter]

11 Feb 2021

PONE-D-20-24388R2 

Effects of therapeutic hypothermia on death among asphyxiated neonates with hypoxic-ischemic encephalopathy: a systematic review and meta-analysis of randomized control trials 

Dear Dr. Abate:

I'm pleased to inform you that your manuscript has been deemed suitable for publication in PLOS ONE. Congratulations! Your manuscript is now with our production department. 

Kind regards, 

on behalf of

Dr. Georg M. Schmölzer 

Academic Editor

PLOS ONE